# Bold Frogs or Shy Toads? How Did the COVID-19 Closure of Zoological Organisations Affect Amphibian Activity?

**DOI:** 10.3390/ani11071982

**Published:** 2021-07-02

**Authors:** Jack Boultwood, Michelle O’Brien, Paul Rose

**Affiliations:** 1Wildfowl & Wetlands Trust, Slimbridge, Gloucestershire GL2 7BT, UK; jack.boultwood@wwt.org.uk (J.B.); michelle.obrien@wwt.org.uk (M.O.); 2Centre for Research in Animal Behaviour, Psychology, Washington Singer, University of Exeter, Exeter EX4 4QG, UK

**Keywords:** amphibian welfare, amphibian behaviour, visitor effect, COVID-19, enclosure usage, evidence-based husbandry

## Abstract

**Simple Summary:**

The visitor effect describes how zoo animals respond to the presence of visitors to their enclosures in a positive, negative or neutral manner. The period of enforced closure of zoos due to the COVID-19 pandemic in March 2020 (and subsequent reopening later that year) allowed for the effect of visitor presence to be evaluated on several species of amphibian. Results have shown that amphibian visibility (i.e., likelihood to be on show in their enclosure) is potentially influenced by the presence of people, and therefore, enclosure layout, collection planning and amphibian husbandry should consider how to minimize any negative influences of the viewing public.

**Abstract:**

Amphibians are an understudied group in the zoo-focussed literature. Whilst commonly housed in specialist exhibits and of real conservation value due to the global extinction crisis, amphibian welfare is not often investigated empirically in zoo settings. The limited research that is available suggests that enclosure design (structure, planting and naturalistic theming) has a positive impact on the time that amphibians will be on show to visitors. However, the categorisation of any “visitor effect” (i.e., influences of visitor presence on amphibian activity and time on display) is hard to find. The COVID-19 pandemic forced the closure of zoological organisations in the UK for several months from March 2020, with gradual re-openings from the summer into autumn and winter. This event provided a unique opportunity to study the effect of the lack of visitors, the presence of essential zoo staff only, the wider return of organisational staff, and then the return of visitors over a prolonged period. This project at WWT Slimbridge Wetlands Centre assessed the number of individuals of six species of amphibian—common toad (*Bufo bufo*), common frog (*Rana temporaria*), smooth newt (*Lissotriton vulgaris*), pool frog (*Pelophylax lessonae*), golden mantella (*Mantella aurantiaca*) and golden poison dart frog (*Phyllobates terribilis*)—visible to observers under different conditions. All amphibians were housed in a purpose-built indoor exhibit of individual enclosures and were recorded when visible (as a proportion of the total population of the enclosure) during closure, the return of extra centre staff and visitor periods. The results showed species-specific differences in visibility, with some species of amphibian being more likely to be on view when the presence of people at their enclosure was less likely or in smaller numbers. Such differences are likely related to the specific camouflage or anti-predation tactics in these focal species. Further study to quantify amphibian sensitivity to, and perception of, environmental change caused by public presence (e.g., light levels and sound) would be useful welfare-themed research extensions. Our results can help inform husbandry, collection planning and amphibian enclosure design to reduce any noticeable visitor effects, and provide a useful benchmark for further, more complex, welfare assessment measures.

## 1. Introduction

The importance of measuring, understanding, and assessing animal welfare has become a focus and priority in recent years for the modern zoological collection. All zoo- and aquarium-housed species (here after “zoo species”) benefit from welfare measurement and assessment as the results from such initiatives can help inform species-relevant decisions related to the animal’s husbandry and management [1]. Bias in the study of zoo species towards mammals can mean less attention is provided to other species [2], and consequently, husbandry may not be evaluated as frequently (or have been evaluated at all) for many bird, reptile, amphibian and invertebrate species in the zoo [3]. Our gap in understanding behaviours, environmental interactions and enclosure use of amphibians, when compared to mammals, may contribute to the low study rate of welfare assessments related to these species [4], as they can be understudied in the zoo [3,5]. The low availability of welfare assessment methods (and their outputs) for amphibians leaves opportunities for positive welfare to be undiscovered, poorer welfare situations to be unrectified and advances to husbandry to be potentially hidden [6]. Amphibians can be well known for their cryptic behaviours, unusual evolutionary traits, and distinctive life history strategies [7]; creating a working one-size-fits-all welfare assessment is unlikely to be successful due to the variation in biological processes and behavioural diversity between individual animals and taxonomic groups [8]. Increasing evidence available for captive care will help to improve the welfare of amphibians in captivity [9,10]. Due to a poor understanding of many amphibian behaviours and their wider natural history, amphibian welfare assessments are often based on an individual’s physical health status and reproductive output, as well as the suitability of environmental parameters, provision of natural furnishings to suggest the absence of poor welfare [2,11] and an opportunity for assumed normal behaviour patterns [10].

These assessments presume that captive amphibians are using the enclosure fully to take advantage of such environmental parameters and enclosure furnishings. Captive amphibian environments can often be replicated to a near natural or functional equivalent of their native habitat. As full habitat replication is not something that is often possible in non-amphibian captive environments, many amphibian species may be able to experience their full behavioural range in such a recreated captive environment [12]. The replication of a natural habitat encourages the performance of adaptive behaviours, and this enhanced behavioural repertoire improves animal welfare [9,13,14]. If amphibians are unable to use the full range of habitat zones within their captive environment, behavioural performance may be restricted, and welfare and health could be negatively affected.

When natural behaviours are viewed by a zoo’s visitors, enclosure dwell times increase and positive perceptions of the animal are more likely [15]; naturalistic enclosure design is noted as enhancing the behavioural repertoire of blue poison dart frogs, *Dendrobates tinctorius azureus* [9] and this has implications for engaging visitor interest in the zoo. Providing evidence-based environments in the zoo improves animal welfare, which in turn improves visitor perception and helps to create a lasting connection or more memorable experience from the visitor’s perspective [16,17]. Such lasting connections could help support pro-conservation or pro-sustainable human behaviour change, as well as encourage repeat visits back to the zoo, thus improving income generation for conservation and education programmes.

An overlooked area of amphibian wellbeing in the zoo is the presence of visitors and their potential effects on behaviour and welfare. Visitor presence has been recorded as influencing enclosure usage in other taxa [18], but any “visitor effect” needs to be measured and evaluated alongside other environmental influences occurring at the same time, e.g., weather conditions [19,20]. Visitor presence may reduce the amount of usable space in the captive environment and consequently reduce the animal’s ability to perform a full behavioural repertoire [21]. Visitor presence can have a mixed effect on animals in zoos, ranging from potentially negative impacts [22], to neutral impacts or no effect [23] and positive influences on animal activity [24]. This mixture of responses to zoo visitors means any “visitor effect” on a specific zoo animal must be assessed on a case-by-case basis [25] if results are to be meaningful. To provide information on how strong the visitor effect may be, for specific individuals and species, periods of time when the zoo is quiet or when visitor numbers are reduced, are helpful to enable the collection of information on baseline activity and enclosure use.

In March 2020, all zoological institutions in the United Kingdom were ordered to close to visitors and non-essential staff due to the COVID-19 pandemic. Animal care staff and other essential workers were the only human interaction most animals in zoos experienced until mid-June, when restrictions slowly lifted. These restrictions offered a unique opportunity to record enclosure usage with zero visitor presence over an extended and continuous period, the slow lifting of restrictions also allowed for a gradual reintroduction and recording of increased visitor presence—in the form of the return of some non-essential staff and then eventually a restricted number of visitors. Consequently, this study was designed to determine the effect of an absence of visitors on the visibility of different species of amphibian housed in a purpose-built exhibit at a conservation-themed visitor attraction in the UK. Due to restriction on the number of people allowed to travel and attend work in person, WWT’s amphibian keepers and a veterinary surgeon collected all data, after receiving training with a standardised method. Methods were designed to be simple and easy to implement around daily work schedules.

## 2. Materials and Methods

The Wildfowl and Wetland Trust (WWT) Slimbridge, Gloucestershire, UK, experienced closures and visitor restrictions due to the COVID-19 pandemic. WWT Slimbridge includes a purpose-built indoor amphibian exhibit called “Toad Hall”, which displayed 14 different amphibian enclosures (as of the time of observation) that hold a range of UK native species and exotic, sub-tropical and tropical amphibian species. All enclosures in “Toad Hall” were designed to provide naturalistic habitats for the specific amphibian species housed in them. This study aimed to record the enclosure usage of several species housed within “Toad Hall” and observational data were collected simultaneously for these populations at opportunistic times throughout the various COVID-19 restrictions. Recording enclosure usage throughout this period should provide information useful to the assessment of amphibian welfare within a captive environment. A null hypothesis that there would be no difference in the visibility of the amphibians during lockdown and in the periods when people returned to WWT Slimbridge was formulated as the basis for further analysis.

### 2.1. Sample Populations

Four enclosures were chosen for this study to provide an opportunity to study any visitor effect on camouflaged (e.g., cryptic body colouring against the environmental background), aposematic (e.g., the display of bright, warning colours that indicate toxicity), aquatic (evolved for a life predominantly underwater) and terrestrial (evolved for a life predominately spent on land) species. “Toad Hall” is located on the ground floor of the main visitor centre at WWT Slimbridge and is approximately 350 m^2^ in area. The journey through “Toad Hall” takes the visitor from a wetland-themed indoor soft play area, past a curved wall of vivaria holding the different amphibian species, to an area set with chairs and tables for drawing and artwork, past access to a classroom and cinema, and finally to the entrance for the outdoor waterfowl collection. Visitors are able to walk past a range of different vivaria in either direction (although for re-opening post 2020 lockdown, a one-way system was in place). A dedicated area for animal care staff is located behind the amphibian displays.

Table 1 provides details of the sample populations for this project. Population sizes remained the same for the duration of the study, as did any mixing of amphibians within exhibits. Ratios of cover (as defined as areas of the exhibit enclosed by vegetation, rockwork or furnishings) to open space (e.g., ground open to the light or open water with no planting) were estimated as percentages of the total surface area of the enclosure. Land (e.g., terrestrial environments) to water (e.g., pools for swimming and submerging) percentages were also estimated as the total of all available area. All amphibians originated from captive-bred stock and were in their adult life stages. The sexing of individual animals was not possible during observational data collection.

### 2.2. Species Ecology

The following species were chosen for data collection due to their differing ecology and behaviour, and therefore, potential sensitivity to public presence:

Smooth newt (*Lissotriton vulgaris*): It is naturally distributed across most of Eurasia but has become invasive to some countries and is considered of Least Concern by the IUCN [26]. Smooth newts are terrestrial for most of the year except during the breeding season, where they return to water. Smooth newts utilise camouflage as a form of protection and inhabit damp meadows and fields, gardens and woodland, most often containing or near to a water source [27].

Common frog (*Rana temporaria*): They are found throughout Europe and considered of Least Concern on the IUCN Red List [28]; they are semi-aquatic and utilise camouflage as a form of protection from predators. Common frogs hibernate during the coldest months within their distribution and emerge when conditions begin to warm for the breeding season. The long hind limbs of the common frog allow them to jump comparatively large distances and evade predation [29].

Common toads (*Bufo bufo*): They are found throughout most of Europe and are considered of Least Concern on the IUCN Red List [30]. Other than during the breeding season, this species is mostly terrestrial, and it has evolved skin pigmentation that provides camouflage from predators. The toad’s natural habitat consists of mixed woodlands, fields, gardens and other damp locations [31].

Pool frogs (*Pelophylax lessonae*): They range across a large portion of Europe and are listed as of Least Concern on the IUCN Red List [32]. This species is often found in bodies of water and can reside in dense vegetation around pool edges. A camouflaged species, their skin pigments of browns, yellows, blacks and greens helps them to blend into their immediate surroundings. During the breeding season, male pool frogs push air through their air sacs to create a loud vibrating sound to attract a mate [33].

Golden (poison) dart frog (*Phyllobates terribilis*): They are endemic to humid forests of Colombia and listed as Endangered on the IUCN Red List [34]. Golden dart frogs are usually bright yellow in colour, but other colours are described, dependent on locality. This bright colour warns predators of their toxicity and this aposematic colouration gives these frogs the ability to sit out in the open rather than hide away like camouflaged amphibian species [35].

Golden mantella (*Mantella aurantiaca*): This is a terrestrial frog endemic to eastern Madagascar, with a restricted distribution, and considered Endangered on the IUCN Red List [36]. This mantella inhabit forests around lentic (i.e., still freshwater) wetlands. The bright orange pigment of the golden mantella’s skin suggests an aposematic strategy to warn predators of their toxicity but they do not freely sit out in the open like other aposematic species. Due to their small size, golden mantella take advantage of small areas of refuge and will often use this to hide away as well as sitting in the open. Wild golden mantella are known to be seasonally active, hiding away in the cooler months and becoming active when the local climate turns warmer [37].

### 2.3. Amphibian Husbandry

All amphibian enclosures were serviced before the 09:30 opening time of the Wetland Centre and this continued throughout COVID-19 restrictions. The Native Species Exhibit and Golden Mantella Exhibit had water areas that were non-filtered, and this required daily visual assessment and frequent water changes depending on the condition of the water. The Pool Frog Exhibit and Dart Frog Exhibit had filtered water areas, and fresh water was added weekly. Water tests were conducted bi-weekly and water changes were performed in response to the results of water testing.

The feed schedules for these exhibits were as follows: Monday: Native Species Exhibit and Pool Frog Exhibit were fed standard house brown crickets. Thursday and Saturday: Golden Mantella Exhibit and Dart Frog Exhibit were fed first banded crickets. All animals were occasionally given an additional feed throughout the week depending on the body condition of the individuals. Spot cleans within the enclosure were performed before feeds and/or before water changes/top ups. Light pruning of growing plants was performed when required to maintain appropriate amphibian refuges and enclosure theming.

The photoperiod for all enclosures was maintained at a 11:13 (day:night) cycle throughout the study, with daylight starting at 08:00 and ending at 19:00. The external lighting in “Toad Hall” was turned on at 08:30 and switched off at 18:00. Each enclosure contained the same number of bulbs that replicated important elements of the light spectrum essential to amphibian health and wellbeing. Each enclosure’s temperature (°C) was controlled by its own heating equipment, regulated by thermostats. Heating was provided along a thermal gradient, with basking spots available. Temperature details for each enclosure for each month of observation are provided in Appendix A Appendix A.

### 2.4. Observational Methods and Sampling Schedule

Data were collected from 24 March 2020 to 18 September 2020 and encompassed periods of complete lockdown (only zoo staff present), partial reopening to other members of the workforce (e.g., cleaning team) on 1 June 2020, and when visitors were allowed back into indoor areas of the attraction, on 1 August 2020. Due to restrictions on staffing number and staff holiday, no data were collected over July 2020.

Sampling was opportunistic around the working schedule of the keepers. Ideally, the presence of amphibians visible and on show (from the public viewing areas of the exhibit) was counted multiple times per day between the arrival of keepers at 08:00 and when they left at 16:30. Using a standardised recording sheet, the observer would stand in the public viewing area of “Toad Hall”, approximately 1 m in front of each vivarium and count the number of individuals that could be seen in that enclosure (and therefore classed as “on show”) before moving to the next vivarium. The observer spent approximately 30 s at each enclosure to scan the environment for visible amphibians and to record how many were on show. No count was recorded if an animal or population was not visible at the time of observation from the public viewing area, and observations were conducted as if the exhibit was being viewed by visitors. All individual animals in all populations were accounted for members of the amphibian care staff each day during daily visual health checks.

An ethical review of the observational method was conducted by the veterinary team at WWT Slimbridge on 23 March; due to COVID restrictions, and the furlough of staff, limited reviewers were available. The project was observational, with no changes to husbandry and no manipulation of animals. Amphibian husbandry remained at the same standard during the lockdown and beyond as per pre-COVID-19 management.

### 2.5. Data Analysis

Data were analysed in RStudio v. 1.4.1106 [38], using R statistical software v.4.0.2 [39] and in Minitab v. 19 [40]. A total of 318 times of observation were conducted, across 128 days of study and 5087 records of the animals were provided. The median number of recordings per day of observation was three (minimum one, maximum five). The median number of daily observations for each month of study was from two to three. The total population visible across all four exhibits was 43 individuals. The median number of individual visible for the entire duration of the study was 16 (Q1: 13, Q3: 19) and the maximum ever seen was 29.

A total of 165 observations (63 days) were made in the period from 24 March (keepers only) to other workers returning to the centre, 59 observations (23 days) during the period of keeper staff plus other members of the workforce and then 94 observations (42 days) from the period from 1 August to the end of the project when staff and visitors were present.

Times of observation were coded into eight hour-long periods based on when the observations fell. These were coded as the top of the hour that the observation started (i.e., time period 8 means observations starts in the hour commencing 08:00). Coded times were: 8 (early morning, keepers arriving), 9 (around normal opening to visitor times), 10 and 11 (mid-morning), 12 and 14 (midday), 15 (early afternoon, post lunch) and 16 (late afternoon, departure of keepers and no more visitor admissions). Periods of data collection were classified as “lockdown” (24 March to 31 May), “cleaners” (1 to 30 June) and “visitors” (1 August to 18 September).

To understand any potential environmental influences on amphibian visibility, temperature records were obtained from the internal thermometer readings of each enclosure (see Appendix A Appendix A). Preliminary Kruskal Wallis tests were run in Minitab, with daily temperature (non-normally distributed) as the response variable and date as the predictor to assess any significant difference in temperature variation each day of study. Each Kruskal Wallis test returned a non-significant result (H = 126.89; df = 127; *p* = 0.486). As a different number of days of study were undertaken each month, the median temperature per month was calculated for each enclosure and included in a one-factor Chi-squared test, run in Minitab, to further check for any association between each observation month and change in internal enclosure temperature. No significant results for month and change in exhibit temperature were identified (Native Species: χ^2^ = 1.10; df = 5; *p* = 0.962; Pool Frog: χ^2^ = 1.09; df = 5; *p* = 0.955; Dart Frog: χ^2^ = 1.05; df = 5; *p* = 0959, Mantella: χ^2^ = 0.823; df = 5; *p* = 0.976), and therefore, temperature was not included in any predictive modelling.

To determine the effect of closure and different stages of reopening on the visibility of amphibians, a bar chart of the median number of individuals visible during each month of observation was produced. A Poisson regression was run in RStudio, using the monthly total count of visible animals as the output variable to correspond with the different period of observation (lockdown, cleaners, visitors) and exhibit (Native Species Exhibit, Pool Frog Exhibit, Golden Mantella Exhibit, and Dart Frog Exhibit) as the categorical predictors. Tukey’s post hoc testing was run to determine where significant differences lay within each predictor.

The proportion of the total population across all exhibits (N = 43) visible for each time of study across all 318 records was calculated and plotted as a scatter plot with a polynomial trend line fitted to account for fluctuations in these data. This trend line provided the best fit (r^2^ = 21%) to these data.

#### 2.5.1. Assessing Ecological Differences

The proportions of time spent on show for the two species of camouflaged frog (common frog and pool frog, N = 12), the two species of aposematic frog (golden mantella and golden poison dart frog, N = 25) and the common toad (N = 2), as the mainly terrestrial species, were calculated for each month of study and presented as an interval plot, drawn in Minitab v.19.

Using the “lmerTest” package in RStudio, a repeated measures model was run for each combination of amphibians (camouflaged frogs, aposematic frogs and toad) in turn. The proportion of visible animals was the output variable and the coded time of day when the observation occurred and the period of observation (lockdown, cleaners or visitors) were the fixed factors. The date was included as the random factor in this model as observations were repeated on the same animals across the same dates. The “plot(model name)” function and the r^2^ value (calculated using the “MuMin” package in RStudio) were used to determine model fit. Tukey’s post hoc testing was run using the “lsmeans” and “pbkrtest” packages in RStudio and the “anova(model name)” function was run to provide model output information using Satterthwaite’s method.

#### 2.5.2. Comparing Pool Frog Populations

Two samples of pool frogs were included in the study: those in a mixed species enclosure (N = 2) and those in a single-species exhibit (N = 8). For each day of study, the total count of visible animals seen for all observations was divided by the total possible number of visible animals (enclosure sample size * number of times of observation per day) to obtain the daily proportion of animals on view. An interval plot was drawn in Minitab v.19 to illustrate these data.

Using the “lmerTest” package in RStudio, a repeated measures model was run using the proportion of visible frogs (out of the total population) per observation point as the output variable, with the exhibit (mixed species or single species holding), time of observation (coded based on the hour of the day) and the period of study (lockdown, when cleaners had returned, and when visitors had returned) as fixed factors. The date was included as a random factor. Model fit, r^2^ values and post hoc testing was carried out in the same way as described above.

Collinearity was tested in all model using the “car” package in RStudio, with the code “vif(model name)” to calculate a variance inflation factor (VIF). VIFs of below 2 were considered acceptable and evidence of no collinearity. All models returned VIFS of 1.003 to 1.5.

## 3. Results

Across all exhibits observed, amphibian visibility declined for the period when visitors were allowed back into the attraction in August and September, whilst the presence of increased numbers of staff “cleaners” appears to have little effect on animals being on display (Figure 1). The exception to this pattern was the Golden Mantella Exhibit, where more animals are visible in September compared to other periods of the year; however, a drop in the number of visible frogs within this exhibit was noted in the first month of visitor presence (August), in conjunction with the pattern seen in other species/exhibits.

Overall, the Poisson regression showed a significant effect of the exhibit, and month/period on the visibility of amphibians (estimate = 5.81; SE = 0.04; z value = 158.7; *p* < 0.001), with the months of August and September (the visitor period) and March (the start of lockdown) showing reduced counts of visible animals (Table 2). The visibility of amphibians at the end of the complete lockdown period (May) was no different to when extra members of the workforce (“cleaners”) arrived back in June. The r^2^ value for this model was 0.75.

Post hoc testing for differences in visibility by exhibit over the months and periods of observation showed that animals in the Native Species Exhibit were less likely to be on display compared to the Pool Frog Exhibit (estimate = −0.472; SE = 0.04; z ratio = −10.9; *p* < 0.001) and animals in the Dart Frog Exhibit and Golden Mantella Exhibit were more likely to be on display compared to the Native Species Exhibit (Dart Frog estimate = 0.5; SE = 0.04; z ratio = 11.71; *p* < 0.001. Golden mantella estimate = 0.47; SE = 0.04; z ratio = 10.9; *p* < 0.001). There was no significant difference over month and period for visibility between the Pool Frog Exhibit, the Golden Mantella Exhibit and the Dart Frog Exhibit.

Figure 2 suggests that whilst a visitor effect was noted for these frogs once the attraction re-opened to visitors, this waned over time and frogs began to be visible in similar numbers, as seen during the lockdown period and the period of increased staff presence (but no visitors).

### 3.1. Assessing Ecological Differences

Smooth newts were not included in the analyses, as out of all 318 records, one newt out of the total population (N = 4) was observed on only two occasions, both in March 2019 during the main lockdown. Common frogs were included in the analysis even though their visibility was low (64 records out of 318), with the majority of these records (64%) occurring during March and April. All amphibians (camouflaged, aposematic and toad) reduced their visibility in August compared to times of the year when fewer people were around. Aposematic species appeared to markedly increase visibility to rates seen during lockdown a month after visitors had returned (Figure 3).

For each ecologically specific group (camouflaged frogs, aposematic frogs and the common toads), the coded time of day of when the observation occurred and the period of study (lockdown, cleaners, visitors) showed a significant relationship with records of visible amphibians. There was a significant difference in the proportion of aposematic frogs being on show for the different periods (lockdown, cleaners, visitors) of the study (F_2, 117.2_ = 13.76; *p* < 0.001) and for different times of the day (F_7, 222.08_ = 6.12; *p* < 0.001). The r^2^ value for this model was 0.64. Significantly more aposematic frogs were on view when cleaning staff returned compared to when the public returned (estimate = 0.15; SE = 0.03; t ratio = 5.17; *p* < 0.001). Interestingly, there was no difference for aposematic frogs being visible for the lockdown period compared to when visitors returned (estimate = 0.04; SE = 0.02; t ratio = 1.64; *p* = 0.235). More aposematic frogs were visible in late morning and afternoon time periods compared to early morning, for example at 3 p.m. compared to 8 a.m. (estimate = 0.08; SE = 0.017; t ratio = 4.62; *p* = 0.0002) and at 11 a.m. compared to 8 a.m. (estimate = 0.1; SE = 0.022; t ratio = 4.402; *p* = 0.0004).

Likewise, camouflaged frogs showed a significant difference for when more animals were visible based on the period (lockdown, cleaners, visitors) of observation (F_2, 106.7_ = 197.24; *p* < 0.001), and time of day showed a limited significant influence on visibility too (F_7, 255.33_ = 2.39; *p* = 0.021). The r^2^ value for this model was 0.73. Camouflaged frogs were less influenced by time of day compared to aposematic frogs, with only one time period (4 p.m.) returning an increased chance of seeing more camouflaged frogs on show when compared to 8 a.m. (estimate = 0.083; SE = 0.023; t ratio = 3.62; *p* = 0.0084). Camouflaged frogs were more likely to be on show during complete lockdown (estimate = 0.333; SE = 0.02; t ratio = 17.31; *p* < 0.001) and when cleaners returned (estimate = 0.423; SE = 0.03; t ratio = 16.79; *p* < 0.001) compared to when the public were allowed back into the centre.

Common toads again showed a significant difference in the proportion of visible animals for the coded times of observation (F_7, 254.7_ = 3.31; *p* = 0.0022) and for the periods of study (F_2, 124.3_ = 8.91; *p* = 0.0002). The r^2^ value for this model was 0.42. Common toads showed the most significant decline in visibility from lockdown to when more staff returned to the centre (estimate = −0.12; SE = 0.042; t ratio = −2.75; *p* = 0.019) and were most likely to be on view when the centre was in lockdown compared to when visitors returned (estimate = 0.135; SE = 0.035; t ratio = 3.90; *p* = 0.0005).

### 3.2. Comparing Pool Frog Populations

Pool frogs in the single species enclosure were more likely to be visible than those housed in the mixed species enclosure (Figure 4). Pool frogs in the mixed “Native Species Exhibit” lived in a larger enclosure (32% bigger than the single species tank) and a smaller social group (N = 2), but they had a smaller pond (20% of available space) and more cover (55% of available surface area). Both populations of pool frogs showed a decline in visible presence when visitors returned in August and September.

The output from a repeated measures model showed that time of day of the observation did not significantly affect the likelihood of seeing a higher number of pool frogs on show (F_7, 619.93_ = 0.881; *p* = 0.512), but the type of exhibit (F_1, 509.03_ = 212.99; *p* < 0.001) and the period of observation (F_2, 116.79_ = 89.45; *p* < 0.001) did significantly influence the proportion of pool frogs visible. The conditional r^2^ for this model was r^2^ = 0.5. Post hoc testing revealed that pool frogs in the mixed enclosure were less likely to be on show compared to those in the single species exhibit (estimate = −0.275; SE = 0.019; df = 509; t ratio = −14.6; *p* < 0.001) and that across both populations, frogs were more likely to be visible during lockdown (estimate = 0.334; SE = 0.03; df = 121; t ratio = 11.42; *p* < 0.001) and when cleaners had returned (estimate = 0.441; SE = 0.04; df = 119; t ratio = 11.56; *p* < 0.001) compared to when visitors had returned. A significant, if less pronounced increase, in the number of frogs visible when cleaners returned was also noted (estimate = 0.107; SE = 0.04; df = 111; t ratio = 3.04; *p* = 0.008) compared to lockdown.

## 4. Discussion

Our results showed that these amphibians responded differently to the changing presence of people throughout different period of lockdown and other COVID-related restrictions. Results suggested a “visitor effect” may influence some of the British native amphibian species to the point where they could be unsuitable for public display (i.e., due to their lack of visibility throughout the study period), with a notable example of this being the smooth newts in this study, which could not be included due to not being visible for most of the observations. The results also showed that some species and populations have differing habituation periods to the return of visitors. The mitigation of such effects could be provided by alterations to the enclosure (e.g., placement or location of the enclosure within the wider exhibit), how the exhibit is accessed by visitors (e.g., consideration of footfall and crowding) and increasing biologically relevant furnishings and refuge spaces). Assessing the results from individual enclosures, alongside data on ecological differences and evolutionary traits, may help to reduce time spent hiding when visitors increase, potentially improving the conservation and educational value of the exhibit [9]. For example, smooth newts inhabited the mixed Native Species Exhibit for the duration of the study and only one newt, out of a population of four, was observed on two occasions during data collection. Both observations were in March during the initial lockdown. This consistent lack of visibility may suggest that the display of smooth newts needs careful consideration; maximising planting that allows the animal to be visible but still feel “under cover” could be useful for the management of this species moving forward. It may also be explained by the seasonal activity pattern of the smooth newt, which is aquatic in spring and summer, becoming terrestrial in the in the autumn [41], and therefore, this species may have been more difficult to see in the enclosure at later times of the study.

The visibility of amphibians within all exhibits either increased or stayed consistent throughout the periods of full lockdown (March, April, and May), as shown in Figure 1. Overall, the total visibility for all amphibians significantly decreased once visitors returned in August and September, and this may suggest that without visitors, these amphibians felt more comfortable being out of or away from cover. The results of visitor presence influencing individuals’ enclosure usage has been reported for other taxa [18], and therefore, the measurement of time spent within specific enclosure zones (alongside the number of visitors at the enclosure) would be useful. Figure 1 indicates that with the introduction of a small number of people, amphibian visibility maintained the same level or increased, and this may be helpful information for zoos to manage visitor flow and number around amphibian enclosures in the future. An animal’s ability to use more of a biologically appropriate enclosure (i.e., one based around information on species ecology) may indicate an improved or consistent opportunity to perform natural behaviours [42]. These results may suggest that during times of no or low visitor presence, amphibians can choose to use different parts of their exhibit in a different way, potentially enhancing their welfare state.

From the March lockdown into an initial “post visitor” period, a significant reduction in amphibian visibility was noted, with golden mantella being recorded to have the fewest number on show (Figure 1). Low numbers of visible amphibians in March may correspond to “normal” levels of viewing when WWT Slimbridge would be open as normal—the animals were simply unaware that external influences had changed. The initial re-introduction of visitors to “Toad Hall” in August appears to cause significant decreases in amphibian visibility across all observed enclosures (Figure 1) and categories of amphibian ecology (Figure 3). This decrease in visibility indicates that enclosure usage was restricted for all amphibians, regardless of ecological niche or species anti-predatory traits. These amphibians have been provided with an enclosure size and resources to suit their biological and behavioural processes—a decrease in amphibian visibility suggests a reduction in enclosure usage as previously mentioned, and this effectively reduces the size of the enclosure, the availability of enclosure resources and the suitability of the exhibit’s stocking density. The reduction in these resources and the enclosure suitability increases the chance of aggression, reduction in basic biological processes, increased competition, and a reduction in overall welfare standards [8]. The evaluation of enclosure suitability should consider where visitor effect may come from and their potential influence on the inhabitants, and not just the physical measurements of the enclosure; such considerations may improve welfare standards for amphibians [43].

The return of visitors over August and September (even when including the increase in golden mantella visibility), showed a sustained and significant decline in total amphibian visibility when compared the lockdown periods. This variation in amphibian visibility may imply a difference in individual species reactions to long-term visitor presence [25]. Figure 2 suggests the total population of amphibians in “Toad Hall” started to habituate to visitor presence and may have returned to lockdown levels of visibility over time.

### 4.1. Ecological and Evolutionary Explanations for an Amphibian Visitor Effect

Figure 3 shows the difference between ecologically specific groups, split into camouflaged frogs, aposematic frogs, and the common toad. When aposematic frogs’ (golden mantella and golden dart frog) visibility is compared to the visibility of the other two ecological groups, there is a significantly higher visibility for these aposematic frogs. This increased visibility of aposematic frogs shows no pattern with time of day. Any increase in visibility in aposematic species is likely influenced by their evolutionary specific traits that allow them to be bolder due to their colouration as an anti-predatory strategy [44], reducing the need to hide away like camouflaged species [45]. Confirming if highly aposematic species, including other mantella and poison dart frog species, are more likely to be bold (i.e., visible in an enclosure) because of an evolutionary confidence related to their toxicity, or if boldness is influenced by enclosure design and furnishing is a relevant future area of study.

The increase in amphibian visibility in the Golden Mantella Exhibit in September, compared to August (Figure 1), may indicate a short habituation period to visitor presence, and such periods of habituation have also been noted in other studies [21]. Combined with the trend seen in March (Figure 1), golden mantella may recover quicker to new situations. Golden mantella will still use cover despite being aposematic, and this mixture of anti-predator responses may have influenced their reaction to an extended visitor presence; initially hiding from visitor presence (August) and then a sudden increase in visibility (September). Such a quick recovery to visitor presence may make them an appropriate candidate for display in the zoo, as visitor presence may be less challenging for this species to cope with. Aposematic species are assumed to be bolder as an evolutionary consequence of being aposematic [45]; exploring if boldness is passive or intentional would allow for better welfare assessments to be conducted for frogs with this evolutionary strategy.

When animals feel more secure in their enclosure, they may be more likely to be on display rather than hidden. As pool frogs featured in both the Native Species Exhibit and the Pool Frog Exhibit, these two populations were compared (Figure 4). Pool frogs were more likely to be visible in the single species Pool Frog Exhibit when compared proportionately to the mixed species Native Species Exhibit. The Native Species Exhibit had a smaller area of water but a larger area of cover in comparison to the Pool Frog Exhibit. This information, combined with the reduced visibility of pool frogs in the Native Species Exhibit, suggested that adequate areas of water are preferred over cover on land; providing this may, in-turn, reduce stress for the pool frogs and increase their visibility. Suitably sized (i.e., based on wild data if available) aquatic areas may be more biologically relevant to pool frogs and increase the chances of frogs being on show. Our results suggest that pool frogs experience a marked visitor effect (Figure 4) regardless of enclosure type or environmental influence; this should be considered during welfare assessments and enclosure design. Pool frogs may experience a critical visitor presence capacity or a habituation period, where visitor presence has no or a negligible effect [23], and further research may produce insights into improving welfare for exhibited pool frogs.

Common toads (as the mainly terrestrial species) were also likely to be on view under lockdown conditions; the toad’s camouflage is its primary defensive strategy, with toxicity being a secondary defence mechanism [31,46]. Therefore, it may be beneficial to offer more cover and camouflage opportunities for captive common toads, in a way that allows the toads to feel hidden but can still be viewed by visitors. As common toads are often nocturnal in their activity patterns at specific times of the year [47], potentially as an adaptive response to predation [48], the measurement of nocturnal behaviour should be conducted to further illuminate the findings from this diurnal study. It is important to remember that records of “on show or hidden” is a relatively crude inference of welfare, and extending this research area to investigate space use within an enclosure, social dynamics between individuals (e.g., to determine resource usage) and behavioural measures of activity (e.g., measuring behaviour of a specific adaptive function to the animal) would add more context to why the animal is on show or not.

### 4.2. Potential Extraneous Influences on Amphibian Enclosure Usage

The position of the observer and their ability to locate aposematic species over camouflaged or cryptic species needs to be considered when measuring the visibility of amphibians within a vivarium. Camouflaged or cryptic species are, by their nature, going to be harder to locate when compared to aposematic species; this could have influenced the count of some species. The observer’s position needs to be considered and standardising the location of data collection (at each enclosure) will reduce discrepancies between observer counts but may not cover all viewing opportunities that a visitor might experience (when visitors may get very close to the enclosure and look more intently for the animal).

Breeding seasons and temporal rhythms can increase “confidence” in amphibian species when they need to display [49]; this may have occurred in June for the pool frog visibility and in September for the golden mantella visibility. As ectotherms, amphibian activity levels are influenced by seasonal fluctuations in temperature, daylight, humidity and precipitation [49]. Although any temperature differences within this (and all) exhibit was not significant, these environmental parameters should be considered when observing the continuous rise in activity levels seen across all lockdown periods in the Golden Mantella Exhibit. The visibility levels are not consistent, but they do trend with seasonal fluctuations in golden mantella activity [50]. The return of visitors triggered a decrease in the visibility of golden mantella initially; the increase in September may have been caused by an environmental influence on their activity levels, and similar situations have been noted in other studies [19].

Whilst temperature fluctuations may not have been significantly different within each enclosure over time, variation in environmental temperature across each month may have been biologically significant, and consequently have had a cumulative effect on behaviour, increasing the chances of amphibians being visible as the study progressed. Further analysis of behaviour patterns alongside the measurement of environmental parameters would allow this to be evaluated. Sex differences and social grouping may also play a role in individual activity; these should be studied to determine which aspects of social dynamics have played a role in the increased presence of frogs in more open areas of the enclosure. Further study of social influences on activity may be especially useful to further interpretation of differences in pool frog visibility, as the larger group in the single species enclosure may have the opportunity to perform a more diverse behavioural repertoire that means they are more likely to be visible.

Other taxa have benefited from research into the type of stress caused by a response to visitor presence [43,51]. Visitors causing vibration, auditory levels, the physical presence of potential danger, a change in light levels, or sudden movements can cause stressors in animals; further research in these areas would help to determine the types of mitigation that can reduce stress and increase visibility.

## 5. Conclusions

The purpose of this study was to take advantage of the COVID-19 lockdown and associated restrictions to examine the reaction of captive amphibians at a specific zoological collection from a period of no visitor presence to that of substantial visitor presence. Although the reaction of species with aposematic traits suggests that they may be more suited to coping with visitor effects than species with camouflaged traits, all amphibians in the study experienced an initial visitor effect, but habituation periods varied. Any “visitor effect” on these captive amphibians appear to be multifactorial; species- and ecological-specific traits, enclosure type and external environmental characteristics are all likely factors influencing any response to visitor presence, ultimately affecting the chances of amphibians being on display. Providing enclosure design based on ecological evidence, as well as a choice of resources, furnishings and refugia, can help mitigate any negative effects of visitor presence. A lack of provision of these enclosure features may compound or increase the negative effects of visitors, and this should be further studied. Internal biological rhythms and seasonal changes, such as reproductive activity, may override the response to visitor presence in some species, but any welfare implications are still unknown. Understanding more about species-specific responses to captive (environmental) influences will provide further evidence for how to provide best practice care for these species of amphibian in the future.

## Figures and Tables

**Figure 1 animals-11-01982-f001:**
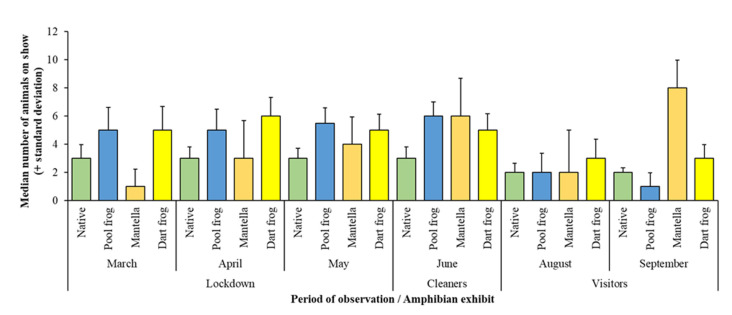
The median number of animals from each exhibit under observation for each of the three study periods (lockdown, when cleaning staff returned, and when visitors returned). Decreases in median number of animals on show is noted for all exhibits except for the Golden Mantella Exhibit.

**Figure 2 animals-11-01982-f002:**
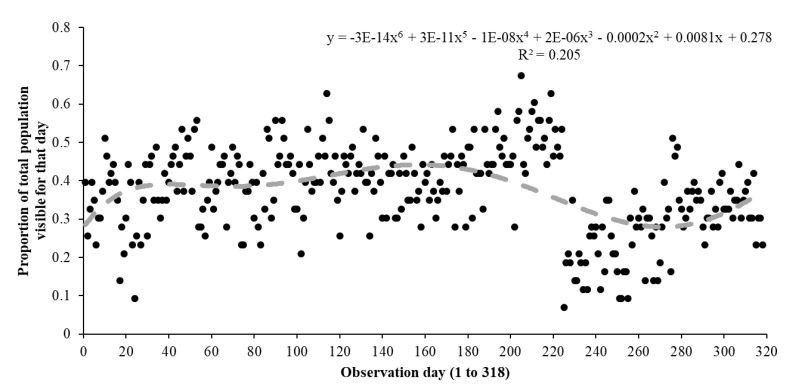
Scatterplot with best fit polynomial trend line showing that amphibian visibility increased during lockdown (day 1 to 165) and then dropped when people returned to the centre (day 166 onwards), but visibility increased again at the end of the study, suggesting the visitor effect may be temporary.

**Figure 3 animals-11-01982-f003:**
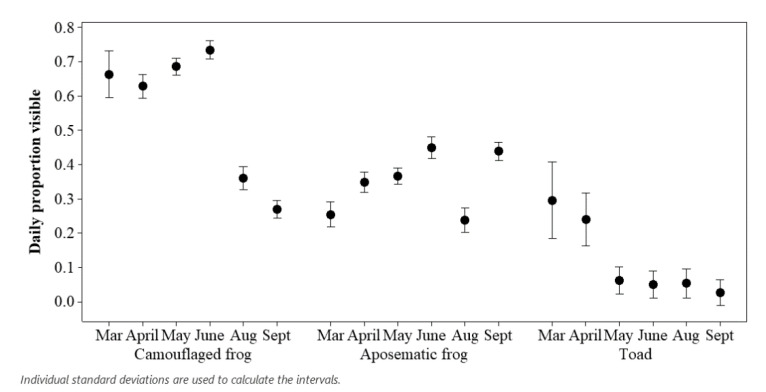
The proportion of camouflaged frogs (common frog and pool frog), aposematic frogs (golden mantella and poison dart frog) and common toad on show for each month of study.

**Figure 4 animals-11-01982-f004:**
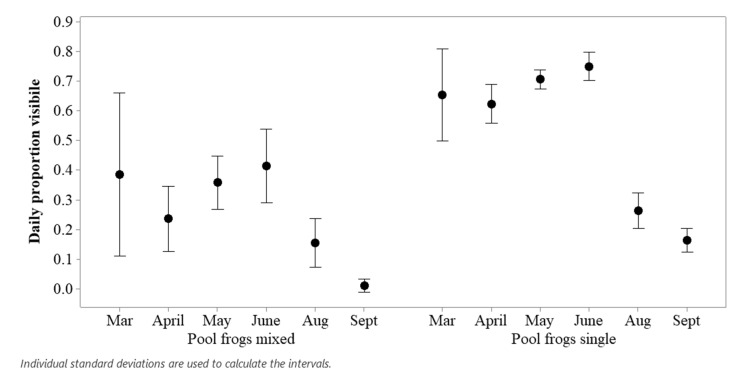
Interval plot of monthly sightings of pool frogs in two exhibits based on the proportion of the total number of potentially visible frogs.

**Table 1 animals-11-01982-t001:** Sample populations and housing details of amphibian observed in this study.

Exhibit	Species Housed	Population	Enclosure Size	% Cover to Open Space	% Land to Water
**Native** **Species** **Exhibit**	Common toad (*Bufo bufo*) Common frog (*Rana temporaria*) Pool frog (*Pelophylax lessonae*) Smooth newt (*Lissotriton vulgaris*)	2 2 2 4	L 130 cm H 50 cm D 50 cm	55%/45%	80%/20%
**Pool Frog Exhibit**	Pool frog	8	L 100 cm H 50 cm D 50 cm	40%/60%	15%/85%
**Golden** **Mantella Exhibit**	Golden mantella (*Mantella aurantiaca*)	18	L 100 cm H 50 cm D 50 cm	65%/35%	90%/10%
**Dart Frog Exhibit**	Golden poison dart frog (*Phyllobates terribilis*)	7	L 100 cm H 50 cm D 50 cm	45%/55%	70%/30%

**Table 2 animals-11-01982-t002:** Model outputs for Poisson regression for start and end of lockdown, period of time when staff returned and the two months when visitors were present.

Predictor	Estimate	Standard Error	Z Value	*p* Value
Lockdown (March)	−1.28	0.06	−20.6	<0.001
Lockdown (May)	0.06	0.04	1.45	0.146
Cleaners (June)	0.001	0.04	0.021	0.984
Visitors (August)	−0.71	0.05	−14.1	<0.001
Visitors (September)	−0.78	0.05	−15.12	<0.001

## Data Availability

Data are available upon reasonable request from the corresponding authors.

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
