# Peer review of "Bold Frogs or Shy Toads? How Did the COVID-19 Closure of Zoological Organisations Affect Amphibian Activity?"

_animals, 2021, doi:10.3390/ani11071982_

Round 1

Reviewer 1 Report

This is a well written, well researched and topical piece of work that has great value to the industry. With the situation being faced due to the pandemic it is good to see the opportunity to investigate the effects of lockdown and reopening on alternative taxa too. The introduction sets the theme of the work, backed up with relevant papers and clearly outlines the need for this work. The method is clearly explained though could be improved by:

  1. providing additional background to the individuals, were these wild caught or captive bred animals, could their ages/sex made a difference to the results?
  2. The data collection process while opportunistic has been standardised though some information as to the duration of observation and exact location (distance from exhibit) should be stated.
  3. What assumptions have been tested to ascertain test such as Tukey test is appropriate?
  4. The categories "camouflaged", "aposematic", "aquatic" and "terrestrial" are introduced at the start of the methods but then change to -aquatic frogs, toads and aposematic later.  Consistency would help to avoid confusion.  Why lose the camouflaged category?

Results have been tested and reported well but could be made easier to read by providing model statistics in a table.

Results have been interpreted and discussed well in line with current research and alternative possibilities for these results have been considered alongside some limitations. 

Overall a very nice piece.

Author Response

Replies to reviewer 1

This is a well written, well researched and topical piece of work that has great value to the industry. With the situation being faced due to the pandemic it is good to see the opportunity to investigate the effects of lockdown and reopening on alternative taxa too. The introduction sets the theme of the work, backed up with relevant papers and clearly outlines the need for this work. The method is clearly explained though could be improved by:

Thank you for the kind words on our paper.

providing additional background to the individuals, were these wild caught or captive bred animals, could their ages/sex made a difference to the results?

Thank you for the comment. These details have been included in the manuscript.

The data collection process while opportunistic has been standardised though some information as to the duration of observation and exact location (distance from exhibit) should be stated.

Thank you for the comment. The approximate distance from the exhibit was been included in the the original manuscript (1m from the enclosure). Time for the observation to run has been included too.

What assumptions have been tested to ascertain test such as Tukey test is appropriate?

We check model fit by visualising plots of standardised residuals before any further testing was conducted.

The categories "camouflaged", "aposematic", "aquatic" and "terrestrial" are introduced at the start of the methods but then change to -aquatic frogs, toads and aposematic later.  Consistency would help to avoid confusion.  Why lose the camouflaged category?

Thank you for the comment. We have clarified our terms in the results section to ensure we match descriptors of the ecology of the animals.

Results have been tested and reported well but could be made easier to read by providing model statistics in a table.

Thank you for the comment. We have included the model output for the poisson regression in a new table. It is harder for the other model outputs because of differences in numerous significant predictors so we would prefer to explain key outputs as text rather than have many large, unwieldy tables, is that is acceptable?

Results have been interpreted and discussed well in line with current research and alternative possibilities for these results have been considered alongside some limitations.

Thank you. we have explained several more limitations to strengthen this.

Overall a very nice piece.

Reviewer 2 Report

The authors have addressed a much-neglected group in this study, and this is enormously welcome. It is particularly laudible that keeping an vet staff working under the enormous time pressure of lockdown took the time to collect the data used in the study.

Although the authors provide evidence of some potentially interesting trends, I have some concerns about the degree to which causality can be attributed to them, and some of the approaches that have been taken in data analysis and interpretation.

The authors group the species into ecological groups in order to compare and contrast visitor effects between these groups. The categories chosen, however, don't represent real ecological groups from the same set. The categories of aposematic/non-aposematic, aquatic/terrestrial and toad are not mutally exclusive and the analyses are therefore comparing apples and oranges; although there are some significant effects, these don't, in my opinion, have any real ecological meaning. The only exception may be aposematic vs. cryptic; aquatic vs. terrestrial is not useful as there are no aquatic, aposematic species to balance the comparison. This issue can be addressed by redefining groups and/or removing some analyses entirely. In addition, there are actually no truly aquatic frogs in the study; both pool and common frogs routinely use both aquatic and terrestrial habitats (the authors themselves note this when discussing the role of terrestrial cover when frogs were not visible). 

The main overall point made by the authors is that changes in visibility are attributable to visitor effects. Although strong coincidence is demonstrated by the results, I do not feel that confounding variables have been appropriately controlled and so causality cannot be established. The authors note that seasonality may have contributed to trends, but I believe that this is a much more fundamental problem with the data analyses and interpretation and one that may limit the validity of the study. The majority of taxa are temperate or subtropical species that are known to have strongly seasonal changes in enclosure use and activity, and/or behavioural responses to environmental temperature. Not only are the increases in visibility of some animals at the end of the study possibly caused by seasonality, as the authors admit, but almost all the trends may be explained in this light. The authors include only one replicate of each lockdown phase (zoos were closed again after the summer period of opening and then opened again, which would have provided replicates of each phase in different seasons), include no temperature/photoperiod/reproductive behaviour or mophology data, and no control using permanently off show animals, and so it is impossible to disentangle visitor and seasonality effects on visibility. I am not sure that this limitation can be overcome at this point, and I feel it may fatally weaken the study.

In addition to these key points, there are several other thematic points that I believe should be addressed in the MS. 

'Visibility' is essentially synonymised with 'higher welfare' throughout the MS, when there is no supporting evidence to indicate that this is the case. Amphibians may respond to stressors with increased or decreased activity and, although the interepreation used byt he authors is more plausible in the circumstances, this assumption should not be made without supporting behavioural evidence. Additionally, the authors repeatedly refer to enclosure usage, and at times equate higher visibility with higher enclosure usage. This was not measured in the study and so this is, I believe, a false equation. 

The authors do not discuss the highly ecologically relevant factor of activity patterns and the fact that for many of the taxa included, the real activity happens after dark. Although visitors are never present at night, the presence of a potential stressor during the day may cause adjusted activity after dark. This may have been difficult data to collect, but certainly worth discussion.

The authors refer specifically to the potential impacts of sound in their abstract, but this variable was not recorded and is one of a multitude of potential mechanisms for any visitor effects. This is therefore misleading, without any supporting evidence, and should be removed. 

The authors exclude a dataset from analyses due to insufficient sample size (smooth newts) but then rely on anecdotal interpretation of this excluded data set to support points made in the Discussion; although a useful side-point, such observations should not underpin the Discussion.

Sex ratios of animals are also not considered, especially in the comparison between the two pool frog exhibits.

In addition to the above concerns, I feel that the MS (although methodically and comprehensively written) could benefit from substantial editing to make it more concise. Some aspects, such as the description of the wild habitat of each species, could be either refined to a line or two, or entirely removed as the information included is not really referred to elsewhere in the MS. There are also several typos, and some sentences that do not make sense due to missing words.

I have added post-it comments on the MS itself, covering some fo the above points and additional smaller notes.

Overall, I have substantial concerns over methodology and analyses, which may or may not be mitigatable. Are temperature and historic reproduction records available for the period, which could be incorporated into models to at least partly control for seasonality, for example? At present I cannot recommend this work for acceptance; i have recommended major revisions (rather than rejection) as if additional datasets are available, this would be a really valuable piece of research in a field that is desperate for attention!

Author Response

Replies to reviewer 2

The authors have addressed a much-neglected group in this study, and this is enormously welcome. It is particularly laudible that keeping an vet staff working under the enormous time pressure of lockdown took the time to collect the data used in the study.

Thank you for the kind words about the intentions of this paper.

Although the authors provide evidence of some potentially interesting trends, I have some concerns about the degree to which causality can be attributed to them, and some of the approaches that have been taken in data analysis and interpretation.

The authors group the species into ecological groups in order to compare and contrast visitor effects between these groups. The categories chosen, however, don't represent real ecological groups from the same set. The categories of aposematic/non-aposematic, aquatic/terrestrial and toad are not mutally exclusive and the analyses are therefore comparing apples and oranges; although there are some significant effects, these don't, in my opinion, have any real ecological meaning. The only exception may be aposematic vs. cryptic; aquatic vs. terrestrial is not useful as there are no aquatic, aposematic species to balance the comparison. This issue can be addressed by redefining groups and/or removing some analyses entirely. In addition, there are actually no truly aquatic frogs in the study; both pool and common frogs routinely use both aquatic and terrestrial habitats (the authors themselves note this when discussing the role of terrestrial cover when frogs were not visible). 

Thank you for the comment. We used the term aquatic simply to denote a difference from the mantellas and dart frogs when running the modelling. We have edited and used aposematic and camouflaged throughout the relevant sections of the manuscript. We use the toad as it was toxic and camouflaged so the more terrestrial nature of the species was most appropriate to pull out. 

The main overall point made by the authors is that changes in visibility are attributable to visitor effects. Although strong coincidence is demonstrated by the results, I do not feel that confounding variables have been appropriately controlled and so causality cannot be established. The authors note that seasonality may have contributed to trends, but I believe that this is a much more fundamental problem with the data analyses and interpretation and one that may limit the validity of the study. The majority of taxa are temperate or subtropical species that are known to have strongly seasonal changes in enclosure use and activity, and/or behavioural responses to environmental temperature. Not only are the increases in visibility of some animals at the end of the study possibly caused by seasonality, as the authors admit, but almost all the trends may be explained in this light. The authors include only one replicate of each lockdown phase (zoos were closed again after the summer period of opening and then opened again, which would have provided replicates of each phase in different seasons), include no temperature/photoperiod/reproductive behaviour or mophology data, and no control using permanently off show animals, and so it is impossible to disentangle visitor and seasonality effects on visibility. I am not sure that this limitation can be overcome at this point, and I feel it may fatally weaken the study.

Thank you for the comment. We have records of temperatures from each enclosure but we did not originally include them as there was no significant difference in temperature by month. However, we have now created a table (supplementary information) of temperature data and included the preliminary chi-squared analysis that showed no significant temperature differences as reason for why this is not a predictor in our models.

We have only one replicate because this is a case study. We relate our findings back to these animals, and we then use this to stimulate wider reasearch. We set out to explain the potential effects of Covid lockdown on animals in one zoo and we thoroughly and fluently explain limitations/extensions. Our methods are clear and well described, stating the challenging situation that research was conducted in and why we wished to know the potential responses of these animals.

In addition to these key points, there are several other thematic points that I believe should be addressed in the MS. 

'Visibility' is essentially synonymised with 'higher welfare' throughout the MS, when there is no supporting evidence to indicate that this is the case. Amphibians may respond to stressors with increased or decreased activity and, although the interepreation used byt he authors is more plausible in the circumstances, this assumption should not be made without supporting behavioural evidence. Additionally, the authors repeatedly refer to enclosure usage, and at times equate higher visibility with higher enclosure usage. This was not measured in the study and so this is, I believe, a false equation. 

Thank you for the comment. We have clarified in the discussion that visibility is a blunt tool and we have suggested other ways of extending inferences of welfare around this point.

The authors do not discuss the highly ecologically relevant factor of activity patterns and the fact that for many of the taxa included, the real activity happens after dark. Although visitors are never present at night, the presence of a potential stressor during the day may cause adjusted activity after dark. This may have been difficult data to collect, but certainly worth discussion.

Thank you for the comment. We do explain the ecology of each species in the methods sections. We have included extra references on nocturnal activity and future areas of study moving forward.

The authors refer specifically to the potential impacts of sound in their abstract, but this variable was not recorded and is one of a multitude of potential mechanisms for any visitor effects. This is therefore misleading, without any supporting evidence, and should be removed. 

Thank you for the comment, we do not believe this is misleading, it is simply a suggestion for future research based on comments we received when developing the paper. We have explained this in the abstract and in the discussion. At no point do we say that we measure it.  

The authors exclude a dataset from analyses due to insufficient sample size (smooth newts) but then rely on anecdotal interpretation of this excluded data set to support points made in the Discussion; although a useful side-point, such observations should not underpin the Discussion.

Thank you for the comment. We are being transparent by explaining why we did not include the limited data points on the smooth newt in the analysis. We have explained in the discussion that this enclosure may need developing if needs are consistently hard to observe. And we suggest ways of potentially improving the newt’s visibility at this specific animal collection.

Sex ratios of animals are also not considered, especially in the comparison between the two pool frog exhibits.

Thank you for the comment. We have mentioned that sex differences may influence whether or not the animals were on show in the discussion, and we make specific reference to the pool frogs.

In addition to the above concerns, I feel that the MS (although methodically and comprehensively written) could benefit from substantial editing to make it more concise. Some aspects, such as the description of the wild habitat of each species, could be either refined to a line or two, or entirely removed as the information included is not really referred to elsewhere in the MS. There are also several typos, and some sentences that do not make sense due to missing words.

Thank you for the comment. We have reviewed the typological errors. We have reduced several sections of the manuscript but adding more detail to clarify these reviewer edits have increased some sections in length.

I have added post-it comments on the MS itself, covering some fo the above points and additional smaller notes.

Thank for the extra notes we have corrected them where required. Some comments have been impossible read as they overlap each other but we have done our best to action.

We include the scientific name of the common toad when first mentioned and have checked multiple sources where common toad is simply stated as the preferred popular name.

Notes on the repeated measures model is covered in the data analysis section, where we state that the anova(model name) function is used.

We do not make assumptions about the pool frog pool size. Again, we simply state this for transparency to make our methods repeatable.

We have removed any ambiguity around enclosure usage.

We have included more information on newt behaviour.

We have provided more discussion on temperature and environmental considerations.

We also feel it is unfair to say that we sideline the limitations and extensions, when we have provided a large part of the discussion to further research to build on this study and provide more useful evidence alongside of our findings and to interpret out findings in more details. We have added to this with more information on why we have different levels of visibility for some species.

Overall, I have substantial concerns over methodology and analyses, which may or may not be mitigatable. Are temperature and historic reproduction records available for the period, which could be incorporated into models to at least partly control for seasonality, for example? At present I cannot recommend this work for acceptance; i have recommended major revisions (rather than rejection) as if additional datasets are available, this would be a really valuable piece of research in a field that is desperate for attention!

Thank you for the comment.  We have attempted to explain these factors in the paper, noting this is a small scale study that is specifically focussed on these four enclosures and used a very simple and quick-to-use observational method. Our paper makes clear, throughout, that this is a building block for future work.

Reviewer 3 Report

The paper raises an important issue, i.e. whether visitors affect the visibility of the amphibians in zoo exhibits. The data confirm the "visitor effect" and underline the necessity to consider it when projecting amphibian enclosures. The paper is well written and the analysis is sound, so it deserves publication. I have some questions: 1) where the individuals born in the zoo or did they acclimatize? 2) Could sociability in some species affect their reaction to visitors? 3) Did authors observe effects on night activities?

Author Response

Replies to reviewer 3

The paper raises an important issue, i.e. whether visitors affect the visibility of the amphibians in zoo exhibits. The data confirm the "visitor effect" and underline the necessity to consider it when projecting amphibian enclosures. The paper is well written and the analysis is sound, so it deserves publication. I have some questions: 1) where the individuals born in the zoo or did they acclimatize? 2) Could sociability in some species affect their reaction to visitors? 3) Did authors observe effects on night activities?

Thank you for the kind words on the suitability of our paper.

The individuals were all captive hatched.

Sociability is a useful point and we have included information on this in the discussion. Thank you for the comment.

No nocturnal observations were possible during this observation period, due to the pressures on staff because of the Covid-lockdown. We have considered the relevance of this for research extension.

Round 2

Reviewer 2 Report

Thanks to the authors for their detailed responses to my comments. I am largely happy with these responses and the changes made according to them (see below for a response to each answer). Apologies for the couple of points where I asked for information that had in fact already been provided. 

I do have a couple of additional points to raise regarding theanalysis of temperature records and the discussion of seasonality.

I feel that a bit more discussion about how seasons and lockdown periods may be confounding (irrespective of temperature data, see below) as amphibians respond to numerous abiotic variables that change  seasonally and which can't be controlled in this study (e.g. photoperiods of background natural lighting, and barometric pressure). Although visitors likely have a large effect, seasonal changes can't be disentangled completely. When I mentioned lack of replicates, I meant replicates of lockdown types, not of animal exhibits; as lockdown was eased and then tightend again, there was potential to collect data from two of each lockdown phase, which would have disentangled season and lockdown phase. I understand that data collection was highly constrained by limited resources and so this may not hav ebeen possibly logistically, however. 

I am really pleased that temperature data were available for analysis. I don't feel that the reported analysis is clear, however; the authors say '...to determine any association between each observation month and change in internal enclosure temperature'; did the authors analyse the median temperature data themselves, or a calculated change in temperature between months?

Irrespective of this, looking at the median/SD data in the supplementary materials, I'm not sure that the chi-squared test done on the data is the best analsysis to do. A GLM/Kruskall-Wallis test (depending on normality etc.) testing for effect of month using the raw daily enclosure temperature data might be more appropriate. The supplementary materials show what look like substantial differences between months in average temperatures, with relatively small SD (medians are presented to it's hard to tell for sure!), so I am not convinced that temperatures remained the same across seasons, at least for some exhibits. Please could the authors look at running the suggested analyses to properly investigate this absolutely critical variable? 

If the data still turn out to support no variation in temperature across months, then I am happy with the MS. If there is evidence for variation in temperature then this needs to be worked into analyses.

Responses to answers to original review points:

Although the authors provide evidence....

Thank you for the comment. We used the term aquatic simply to denote a difference from the mantellas and dart frogs when running the modelling. We have edited and used aposematic and camouflaged throughout the relevant sections of the manuscript. We use the toad as it was toxic and camouflaged so the more terrestrial nature of the species was most appropriate to pull out. 

I am happy with this change; I feel that it strengthens the biological meaning of the analyses.

The main overall point made by the authors...

Thank you for the comment. We have records of temperatures from each enclosure but we did not originally include them as there was no significant difference in temperature by month. However, we have now created a table (supplementary information) of temperature data and included the preliminary chi-squared analysis that showed no significant temperature differences as reason for why this is not a predictor in our models.

We have only one replicate because this is a case study. We relate our findings back to these animals, and we then use this to stimulate wider reasearch. We set out to explain the potential effects of Covid lockdown on animals in one zoo and we thoroughly and fluently explain limitations/extensions. Our methods are clear and well described, stating the challenging situation that research was conducted in and why we wished to know the potential responses of these animals.

See above notes on replicates and temperature data.

'Visibility' is essentially synonymised...

Thank you for the comment. We have clarified in the discussion that visibility is a blunt tool and we have suggested other ways of extending inferences of welfare around this point.

I am happy with the adjustments made here, although reference to stressors causing increased activity in some amphibians might be useful to explicitly acknowledge that this may occur.

The authors do not discuss the highly ecologically relevant factor of activity patterns ...

Thank you for the comment. We do explain the ecology of each species in the methods sections. We have included extra references on nocturnal activity and future areas of study moving forward.

Agreed that ecological facts are stated in the Methods; the additional points about impacts on activity outside the observation period strengthen the Discussion. 

The authors refer specifically to the potential impacts of sound in their abstract... 

Thank you for the comment, we do not believe this is misleading, it is simply a suggestion for future research based on comments we received when developing the paper. We have explained this in the abstract and in the discussion. At no point do we say that we measure it.  

My point was that the Abstract specifcally referred to only sound as a potential mechanism for causing visitor effects; the addition of 'e.g.' and other potential mechanisms clarifies that this is hypothetical and avoids accidentally misleading the reader into thinking that sound was actually identified as a cause. Thank you for maing this adjustment.

The authors exclude a dataset from analyses ...

Thank you for the comment. We are being transparent by explaining why we did not include the limited data points on the smooth newt in the analysis. We have explained in the discussion that this enclosure may need developing if needs are consistently hard to observe. And we suggest ways of potentially improving the newt’s visibility at this specific animal collection.

Thank you for making this addition - the transparency in reporting an excluded data set is indeed laudable; I simply wanted to avoid the MS using evidence from the only excluded data set to support a point. The added caveat now addresses this. 

Sex ratios of animals are ...

Thank you for the comment. We have mentioned that sex differences may influence whether or not the animals were on show in the discussion, and we make specific reference to the pool frogs.

The addition makes this poiont well - thank you for making the change.

In addition to the above concerns....

Thank you for the comment. We have reviewed the typological errors. We have reduced several sections of the manuscript but adding more detail to clarify these reviewer edits have increased some sections in length.

This all makes sense - thank you.

I have added post-it comments on the MS itself, covering some fo the above points and additional smaller notes...

Thank for the extra notes we have corrected them where required. Some comments have been impossible read as they overlap each other but we have done our best to action.

It's frustrating that the comments became unreadable after the file was uploaded and formatting changed from a .doc file, where this wouldn't have been a problem.

We include the scientific name of the common toad when first mentioned and have checked multiple sources where common toad is simply stated as the preferred popular name.

This is a very minor point, but in the USA Anaxyrus americanus is called 'common toad', whereas in Asia Duttaphrynus melanostictus is known by the same vernacular name. As you say, the scientific name should clear up any confusion.

Notes on the repeated measures model is covered in the data analysis section, where we state that the anova(model name) function is used.

Accepted - apologies for missing this.

We do not make assumptions about the pool frog pool size. Again, we simply state this for transparency to make our methods repeatable.

Accepted.

We also feel it is unfair to say that we sideline the limitations and extensions, when we have provided a large part of the discussion to further research to build on this study and provide more useful evidence alongside of our findings and to interpret out findings in more details. We have added to this with more information on why we have different levels of visibility for some species.

My primary concern here was that there wa s amajor ncontrolled variable, which was batted aside by the discussion rather than dealt with fully. The new points and the additional analyses address this concern. Thank you for these adjustments.

Overall, I have substantial concerns over methodology and analyses...

Thank you for the comment.  We have attempted to explain these factors in the paper, noting this is a small scale study that is specifically focussed on these four enclosures and used a very simple and quick-to-use observational method. Our paper makes clear, throughout, that this is a building block for future work.

Inclusion of temperature data largely addresses this area (see above comments, though). Indeed, a small scale study, but important potentially confounding variables still need to be controlled where possoble and discussed fully where not. 

Author Response

Replies to reviewer 2

I am really pleased that temperature data were available for analysis. I don't feel that the reported analysis is clear, however; the authors say '...to determine any association between each observation month and change in internal enclosure temperature'; did the authors analyse the median temperature data themselves, or a calculated change in temperature between months?

Irrespective of this, looking at the median/SD data in the supplementary materials, I'm not sure that the chi-squared test done on the data is the best analsysis to do. A GLM/Kruskall-Wallis test (depending on normality etc.) testing for effect of month using the raw daily enclosure temperature data might be more appropriate. The supplementary materials show what look like substantial differences between months in average temperatures, with relatively small SD (medians are presented to it's hard to tell for sure!), so I am not convinced that temperatures remained the same across seasons, at least for some exhibits. Please could the authors look at running the suggested analyses to properly investigate this absolutely critical variable? 

Thank you for the comment. I originally ran a KW test on these data and they all returned non-significance between dates and the temperatures recorded. I asked the amphibian keeper about this as I wanted to check that this was the case in reality and they replied saying that they would expect very similar parameters between the enclosure during the period of time the observations were conducted.

Given that irregular numbers of dates were recorded per enclosure, for ease of presenting these preliminary analyses, I calculated the median (temperature values were not normally distributed) and SD for each month to show that whilst there appears to be some fluctuation, it was not significantly so, and that is where I used a Chi squared test. We did this to show the reader the descriptive analysis prior to inferential testing.

Please see the outputs from the original KW test below.

I am happy to provide copy of the communication between the me and the amphibian keeper that shows the original discussion had about the consistency of temperature as how this lack of variation was to be expected in these tanks during the time of year that the study tool place over.

Native tank

Test

Null hypothesis

H₀: All medians are equal

Alternative hypothesis

H₁: At least one median is different

Method

DF

H-Value

P-Value

Not adjusted for ties

127

126.94

0.485

Adjusted for ties

127

127.00

0.483

The chi-square approximation may not be accurate when some sample sizes are less than 5.

Pool frog tank

Test

Null hypothesis

H₀: All medians are equal

Alternative hypothesis

H₁: At least one median is different

Method

DF

H-Value

P-Value

Not adjusted for ties

127

126.94

0.485

Adjusted for ties

127

127.00

0.483

The chi-square approximation may not be accurate when some sample sizes are less than 5.

Mantella tank

Test

Null hypothesis

H₀: All medians are equal

Alternative hypothesis

H₁: At least one median is different

Method

DF

H-Value

P-Value

Not adjusted for ties

127

126.89

0.486

Adjusted for ties

127

127.00

0.483

The chi-square approximation may not be accurate when some sample sizes are less than 5.

Dart frog tank

Test

Null hypothesis

H₀: All medians are equal

Alternative hypothesis

H₁: At least one median is different

Method

DF

H-Value

P-Value

Not adjusted for ties

127

126.90

0.486

Adjusted for ties

127

127.00

0.483

The chi-square approximation may not be accurate when some sample sizes are less than 5.

We thank the reviewer for their diligence in considering temperature and we would like to allay any fears that we did not consider this. As ectothermic species, all authors are fully aware of how environmental parameters will influence amphibian activity and this was fully recorded and tested prior to further inferential analysis.

I have edited this section of the manuscript accordingly to include the original KW testing and explain that it was followed up with the chi squared test to further check on temperature variation.

I have read over the other comments from the reviewer to the original review and I cannot see any other that need answering but please do get back to me if I have missed any.